# Constraining Gaussian Processes Regression with Quasi-Likelihood Constraint Relaxation

## Abstract

Gaussian Process regression is a popular method for nonparametric, probabilistic modelling. One of its main attractions is also, in some contexts, a significant challenge; namely its high flexibility. This flexibility can be reduced by imposing constraints on the GP prior or posterior, something that there is a large and growing body of literature on. In this paper, we present a generalisation of virtual point methods and a framework for enforcing a broad range of constraints in GP posteriors. The method involves designing a quasi-likelihood function which encodes a relaxed form of the constraints, and then conditioning the unconstrained GP posterior on this quasi-likelihood. The method leverages ideas from existing methods for constrained GP regression, namely Riihimäki and Vehtari (2010) and Hansen et al. (2024), and expands these approaches to a much broader range of constraints. The method is demonstrated with a synthetic example, where a 2-dimensional GP posterior is required to have a divergence-free gradient, as well as real-world example where the posterior GP of Thomson scattering data from the MAST tokamak is required to be both monotonically decreasing and strictly positive.

## 1 Introduction

Gaussian Process (GP) regression is an increasingly popular method for nonparametric, probabilistic modelling. It is an attractive method for several reasons, including its theoretical simplicity, its high flexibility, its high degree of explainability, and its robustness to noisy and uncertain data. Conversely, its drawbacks include its high computational complexity of $\mathcal{O}(n^3)$, space complexity of $\mathcal{O}(n^2)$, the challenge of optimising the hyperparameters. As it turns out, ironically, one of its greatest advantages is also in some cases a significant challenge, namely its high flexibility.

A Gaussian Process is completely specified by its mean $m(\boldsymbol{x})$ and covariance function $k(\boldsymbol{x}, \boldsymbol{x}')$ (Rasmussen and Williams, 2006) for $\boldsymbol{x}, \boldsymbol{x}' \in \mathcal{X}$. If we for a moment disregard the mean function, which is often simply set to zero anyway, the only constraints imposed on the GP prior are those induced by the covariance function. Often these take the form of length scale hyperparameters, which broadly control the smoothness of the permissible functions. While this may appear to be a largely technical fact, it has important implications for the outcomes of GP regression. As such, the covariance function provides a set of *relative* constraints (point-wise function values are constrained only relative to other function values), rather than *absolute* ones. In this context, an "absolute" constraint is taken to mean a constraint pertaining to the function values themselves (and/or their derivatives), such as fixed values, boundedness, monotonicity, etc. It is such "absolute" constraints that we are addressing in this paper, since they provide a means for reducing the flexibility of a GP by tailoring it to a specific application.

In practice, the high flexibility of the GP prior can sometimes lead to underconfident predictions, particularly in data-scarce regimes, where there isn't sufficient data to implicitly propagate relevant constraints to the GP posterior. Additionally, unconstrained GPs may also sometimes produce predictions that are entirely unphysical. For this reason, significant effort has been invested in developing methods for explicitly imposing constraints on GPs. These methods broadly fall into two categories; those that modify the prior GP such that the constraints are somewhat automatically enforced – and those that use additional virtual observations to manually incorporate the constraints in the GP posterior. Imposing constraints by modifying the prior can be done by either warping

the latent GP to conform to the constraints (Jensen et al., 2013), designing a bespoke covariance kernel (see e.g. Wahlström, 2015; Jidling et al., 2017) – something that is only possible for a small subset of constraints – or by fitting the data in a constrained (finite) basis, which effectively leads to a prior that is not a Gaussian Process (see e.g. López-Lopera et al., 2018; Maatouk and Bay, 2017). However, in this work we are mainly concerned with constraining GPs using virtual observations.

More specifically, boundedness can be imposed through a variety of different approaches, such as using warping functions or bounded likelihood functions as explored by Jensen et al. (2013), virtual observations as described in Da Veiga and Marrel (2012), or with an anamorphosis, which is a common approach in climatology literature (Berrocal et al., 2008; Chilès and Delfiner, 2012; Kleiber et al., 2012; 2023). Monotonicity can be enforced by conditioning the GP on virtual observations of the function derivatives under a probit likelihood (Riihimäki and Vehtari, 2010), or by projecting the data onto a basis of monotonic functions and then fit surrogates to the coefficients of the projected data (Pepper et al., 2023). In a similar avenue of enquiry, Maatouk and Bay (2017) outline a framework for enforcing either boundedness or monotonicity using a spline basis. Recently, Hansen et al. (2024) developed a method for softly enforcing physical conservation laws, from which the method presented here borrows some ideas. Section 2.2 contains a more detailed overview of selected related work.

In this paper, we present a novel methodology which can be seen as a generalisation of virtual point methods or, alternatively, as a relaxed form of rejection sampling. It utilises virtual measurements in conjunction with a quasi-likelihood function which encodes the relevant constraints. It can be used to enforce a broad range of constraints, including fixed values, boundedness, monotonicity, integral conservation laws, and incompressibility. We demonstrate the use of the method with a synthetic example where we require the gradient of a 2D surface to be divergence-free, and a real-world example of Thomson scattering data from the field of magnetic confinement fusion, where the data-generating process is expected to be both monotonically decreasing and strictly positive.

## 2 METHODS

In this section, we present the core idea of the proposed methodology. We first briefly recollect the fundamentals of GP modelling in a Bayesian context, including noise heteroskedasticity. We then review some existing methods related to the one presented in this work. Finally, we introduce the proposed quasi-likelihood constraint relaxation Gaussian Process (QLCR-GP) methodology, and give some examples of how to implement common constraints within this framework.

### 2.1 GAUSSIAN PROCESS REGRESSION

Formally, a GP is a collection of random variables $\{f(\boldsymbol{x})|\boldsymbol{x} \in \mathcal{X}\}$ such that any finite set of these random variables $\boldsymbol{f} = \{f(\boldsymbol{x}_i)\}_{i=1}^N$ has a multivariate Gaussian distribution. A GP prior with mean function $m(\boldsymbol{x})$ and covariance function $k(\boldsymbol{x}, \boldsymbol{x}')$ is typically represented using the notation

$$f(\boldsymbol{x}) \sim \mathcal{GP}(m(\boldsymbol{x}), k(\boldsymbol{x}, \boldsymbol{x}')). \tag{1}$$

The prior induced by this GP is conditioned on measurements $\mathcal{D} = \{\boldsymbol{X}, \boldsymbol{y}\}$ to yield the posterior:

$$p(\boldsymbol{f}|\boldsymbol{X}, \boldsymbol{y}) = \frac{p(\boldsymbol{f}|\boldsymbol{X})\,\mathcal{L}(\boldsymbol{y}|\boldsymbol{X}, \boldsymbol{f})}{p(\boldsymbol{y}|\boldsymbol{X})} \tag{2}$$

where $\mathcal{L}(\boldsymbol{y}|\boldsymbol{X}, \boldsymbol{f})$ is the likelihood and $p(\boldsymbol{y}|\boldsymbol{X})$ the the marginal likelihood or *evidence*. Please refer to e.g. Rasmussen and Williams (2006) and Gramacy (2020) for more details.

Typically, the likelihood $\mathcal{L}(\boldsymbol{y}|\boldsymbol{X}, \boldsymbol{f})$ encodes a homoskedastic noise model $\boldsymbol{\epsilon} \sim \mathcal{N}(\boldsymbol{0}, \sigma^2)$, where $\sigma^2$ is the scalar measurement noise variance, which is constant for all $\boldsymbol{x}$. However, in the second example in Section 3.2, we employ a heteroskedastic noise model, where the noise variance is modelled by a second GP, $\log \sigma^2(\boldsymbol{x}) \sim \mathcal{GP}(m_{\sigma^2}(\boldsymbol{x}), k_{\sigma^2}(\boldsymbol{x}, \boldsymbol{x}'))$ (Goldberg et al., 1997; Kersting et al., 2007). This requires measurements of the noise variance, which are available for that example.

## 2.2 RELATED WORK

Constrained GPs is a highly active field of research. Here we review some existing methods related to the one developed here, including fixed value constraints, monotonicity constraints and conservation-law constraints. The method presented here belongs in the family of virtual point methods, rather than methods that modify the prior as seen in e.g. Maatouk and Bay (2017); López-Lopera et al. (2018); Wahlström (2015); Jidling et al. (2017). The following review of related work is therefore mainly concerned with virtual point methods. For a more comprehensive review of constrained GP regression, please refer to Swiler et al. (2020).

### 2.2.1 FIXED VALUE CONSTRAINTS

Fixed value constraints, such as fixed boundary conditions in the context of PDE modelling, can be enforced by placing noise-free virtual observations at the appropriate points. While the observations $\{y_i\}_{i=1}^N$ are usually thought of as measurements, they could also be thought of as observations in the broader sense, namely representative of something that is *known*. Hence, the *authentic* measurements $\mathcal{D}$ can be extended with noise-free *virtual* measurements $\tilde{\mathcal{D}} = \{\tilde{\boldsymbol{x}}_i, \tilde{y}_i\}_{i=1}^M$ representative of the fixed-value constraints to yield an augmented dataset $\mathcal{D}' = \mathcal{D} \cup \tilde{\mathcal{D}}$. The GP is then simply conditioned on the augmented dataset $\mathcal{D}'$ to enforce the constraints. This approach is exact for point-wise fixed value constraints, and can be used to enforce e.g. fixed boundary values for one-dimensional differential equations. However, it is not possible to enforce constraints exactly other than point-wise, and constraints that exist on e.g. a surface can only be enforced approximately by placing sufficiently many virtual points on that surface. Moreover, the required number of points suffers from the curse of dimensionality and will increase exponentially with the input dimension $\dim(\mathcal{X})$. Since GP posteriors are, in a sense, strongly coupled with their training data, this reliance on virtual datapoints is a recurring theme in constrained GP regression.

### 2.2.2 MONOTONICITY CONSTRAINTS

Monotonicity can be enforced by placing a probit likelihood over the derivative of the Gaussian Process at virtual points $\boldsymbol{X}_m$, as explained in Riihimäki and Vehtari (2010). They specify a joint prior over function values and their derivatives by

$$p(\boldsymbol{f}, \boldsymbol{f}'|\boldsymbol{X}, \boldsymbol{X}_m) = \mathcal{N}(\boldsymbol{f}_{\text{joint}}|\boldsymbol{0}, \boldsymbol{K}_{\text{joint}})$$

where

$$\boldsymbol{f}_{\text{joint}} = \begin{bmatrix} \boldsymbol{f} \\ \boldsymbol{f}' \end{bmatrix}, \text{and} \quad \boldsymbol{K}_{\text{joint}} = \begin{bmatrix} \boldsymbol{K}_{\boldsymbol{f},\boldsymbol{f}} & \boldsymbol{K}_{\boldsymbol{f},\boldsymbol{f}'} \\ \boldsymbol{K}_{\boldsymbol{f}',\boldsymbol{f}} & \boldsymbol{K}_{\boldsymbol{f}',\boldsymbol{f}'} \end{bmatrix}$$

where $\boldsymbol{f}'$ denotes the derivative of the function values $\boldsymbol{f}$ with respect to some of the input dimensions. The different varieties of $\boldsymbol{K}$ are the covariance matrices for the variables indicated by the subscript. The joint posterior is then

$$p(\boldsymbol{f}, \boldsymbol{f}'|\boldsymbol{y}, \boldsymbol{m}) \propto p(\boldsymbol{f}, \boldsymbol{f}'|\boldsymbol{X}, \boldsymbol{X}_m)\mathcal{L}(\boldsymbol{y}|\boldsymbol{f})\mathcal{L}(\boldsymbol{m}|\boldsymbol{f}')$$

where the likelihood for the derivative information $\mathcal{L}(\boldsymbol{m}|\boldsymbol{f}')$ is a relaxed probit function. The posterior is intractable because of the non-Gaussian likelihood, and the authors of Riihimäki and Vehtari (2010) use the Expectation Propagation (EP) algorithm to sample from the constrained posterior. Note that an analogous approach can be employed on higher order derivatives to enforce e.g. convexity constraints.

### 2.2.3 CONSERVATION CONSTRAINTS

In a recent paper, Hansen et al. (2024) develop a novel method for enforcing conservation law constraints on Gaussian processes, and other probabilistic predictive models. If $\boldsymbol{G}$ is a matrix approximation to the integral operator that encodes the conservation law and $\boldsymbol{b}$ is a vector of constraint values, the conservation law constraint can be written as the linear equation

$$\boldsymbol{b} = \boldsymbol{G}\boldsymbol{f} + \varepsilon$$

where $\varepsilon \sim \mathcal{N}(\boldsymbol{0}, \nu_G^2 I)$ is a quasi-noise term. The variance $\nu_G^2$ of this quasi-noise can be thought of as a relaxation parameter that expresses how much the conservation constraint can be violated. The unconstrained Gaussian process posterior over test points $\mathcal{D}_{\text{test}} = \{\boldsymbol{x}_i, y_i\}_{i=1}^N$ is Gaussian $\boldsymbol{f} \sim \mathcal{N}(\boldsymbol{\mu}, \boldsymbol{\Sigma})$ with mean $\boldsymbol{\mu}$ and covariance matrix $\boldsymbol{\Sigma}$. This distribution can now be reconditioned on the (discrete) conservation constraint to yield a $\boldsymbol{G}$-constrained Gaussian posterior with mean vector and covariance matrix

$$\boldsymbol{\mu}_G = \boldsymbol{\mu} - \boldsymbol{\Sigma}\boldsymbol{G}^\top(\nu_G^2 I + \boldsymbol{G}\boldsymbol{\Sigma}\boldsymbol{G}^\top)^{-1}(\boldsymbol{G}\boldsymbol{\mu} - \boldsymbol{b}), \tag{3}$$

$$\boldsymbol{\Sigma}_G = \boldsymbol{\Sigma} - \boldsymbol{\Sigma}\boldsymbol{G}^\top(\nu_G^2 I + \boldsymbol{G}\boldsymbol{\Sigma}\boldsymbol{G}^\top)^{-1}\boldsymbol{G}\boldsymbol{\Sigma}. \tag{4}$$

### 2.3 Quasi-Likelihood Constraint Relaxation

In this section, we develop the core idea of this paper. Consider an arbitrary constraint

$$\Phi(f(\boldsymbol{x})) = 0 \tag{5}$$

where $\Phi$ is some functional of an (unconstrained) Gaussian process latent function $f(\boldsymbol{x})$. While such constraints can be enforced exactly, they may introduce a discontinuity in the posterior, which can be challenging to deal with computationally. One common way of doing so is by means of a rejection sampler, which simply filters out any $f(\boldsymbol{x})$ that do not respect the constraint. However, for most types of constraints, this is highly inefficient, since many (or most) samples are rejected. If we instead relax the constraint, we can construct a quasi-likelihood function $\mathcal{L}_\Phi(\Phi|\boldsymbol{f})$, which (broadly speaking) measures the likelihood of the constraint $\Phi$ given an unconstrained Gaussian process latent function $f(\mathbf{x})$. For example, if we place a Gaussian distribution over the constraint residual, we can write that

$$\Phi(f(\boldsymbol{x})) = \varepsilon \quad \text{with} \quad \varepsilon \sim N(\boldsymbol{0}, \nu^2 I) \tag{6}$$

The posterior outlined in Section 2.1 $p(\boldsymbol{f}|\boldsymbol{X}, \boldsymbol{y})$ can now be reconditioned on this quasi-likelihood to yield a $\Phi$-constrained posterior

$$p_\Phi(\boldsymbol{f}|\boldsymbol{X}, \boldsymbol{y}, \Phi) = \frac{p(\boldsymbol{f}|\boldsymbol{X}, \boldsymbol{y})\mathcal{L}_\Phi(\Phi|\boldsymbol{f})}{p(\Phi|\boldsymbol{X}, \boldsymbol{y})}. \tag{7}$$

with $p(\Phi|\boldsymbol{X}, \boldsymbol{y}) = \int \tilde{\mathcal{L}}(\Phi|\boldsymbol{f})p(\boldsymbol{f}|\boldsymbol{X}, \boldsymbol{y}) \, d\boldsymbol{f}$.

This is not unlike the approach taken in Hansen et al. (2024), where the conservation law constraints are also softly enforced through the relaxation parameter $\nu_G^2$. In fact, we can rewrite their conservation constraint as a $\Phi$-constraint as $\Phi(\boldsymbol{u}) = \boldsymbol{G}\boldsymbol{u} - \boldsymbol{b} = \varepsilon$, and we will recover the method of Hansen et al. (2024). Note that when the constraint is linear and the residual is assumed to be Gaussian $\varepsilon \sim \mathcal{N}(\boldsymbol{0}, \nu^2 I)$, as is the case with these conservation constraints, the $\Phi$-constrained posterior is also a Gaussian with an exact solution $p_\Phi(\boldsymbol{f}|\boldsymbol{X}, \boldsymbol{y}, \Phi)$, see Section 2.2.3. For non-linear constraints and/or non-Gaussian residual models, the posterior can be approximated using variational methods or Markov Chain Monte Carlo (MCMC).

#### 2.3.1 Constraints

Here, we list a (non-exhaustive) selection of particular constraints that have been encoded to the form specified in Eq. 5. In the following, consider a set of virtual points in the input space $\{\tilde{\boldsymbol{x}}_i\}_{i=1}^M$, $\tilde{\boldsymbol{x}}_i \in \mathcal{X}$, where the constraints will be enforced.

**Fixed values** Generally, fixed value constraints $\{\tilde{y}_i\}_{i=1}^M$ at locations $\{\tilde{\boldsymbol{x}}_i\}_{i=1}^M$ can be encoded with our QLCR-GP approach by simply setting

$$f(\boldsymbol{x}) - y = \varepsilon \quad \forall \quad \boldsymbol{x}, y \in \{\tilde{\boldsymbol{x}}_i, \tilde{y}_i\}_{i=1}^M \tag{8}$$

This corresponds to adding virtual measurements $\tilde{\mathcal{D}} = \{\tilde{\mathbf{x}}_i, \tilde{y}_i\}_{i=1}^M$, as also explained in Section 2.2.1 to the dataset, and to fix the noise variance of these virtual measurements to $\nu^2$ in the quasi-likelihood. Similarly, for fixed boundary value constraints, for example $f(x, 0) = c$, with $f : \mathbb{R}^2 \to \mathbb{R}$ we can require that

$$f(x, 0) - c = \varepsilon \quad \forall \quad x \in \{\tilde{x}_i\}_{i=1}^M. \tag{9}$$

This is a special case of fixed value constraints, and a common use case in PDE-based modelling.

**Positivity**   For positivity constraints, we can for example require that

$$g(\boldsymbol{x}) = \varepsilon \quad \forall \quad \boldsymbol{x} \in \{\tilde{\boldsymbol{x}}_i\}_{i=1}^M \quad \text{with} \quad g(\boldsymbol{x}) = \begin{cases} f(\boldsymbol{x}), & \text{if } f(\boldsymbol{x}) < 0 \\ 0, & \text{otherwise.} \end{cases} \tag{10}$$

This can be viewed as a type of hinge-loss, where function values that violate the constraints are penalised proportionally to the violation, while function values that do not violate the constraint are ignored. We employ the same hinge-loss idea to enforce monotonicity and convexity in the following.

**Monotonicity**   To constrain $f(x)$ to be monotonically increasing with $f : \mathbb{R} \to \mathbb{R}$, we can require that

$$h(x) = \varepsilon \quad \forall \quad x \in \{\tilde{x}_i\}_{i=1}^M \quad \text{with} \quad h(x) = \begin{cases} \frac{df}{dx}(x), & \text{if } \frac{df}{dx}(x) < 0 \\ 0, & \text{otherwise.} \end{cases} \tag{11}$$

This is equivalent of Eq. 10, but it concerns the latent function derivative $\frac{df}{dx}(x)$ rather than the function. This could also easily be extended to convexity-constraints by choosing

$$h(x) = \begin{cases} \frac{d^2 f}{dx^2}(x), & \text{if } \frac{d^2 f}{dx^2}(x) < 0 \\ 0, & \text{otherwise.} \end{cases} \tag{12}$$

Note that the derivative of a GP is also a GP since differentiation is a linear operation (Solak et al., 2002; Rasmussen and Williams, 2006; Riihimäki and Vehtari, 2010), and thus the function $f(x)$ and its derivative $\frac{df}{dx}(x)$ – and second derivative $\frac{d^2 f}{dx^2}(x)$ – can be modelled jointly using a GP with an appropriate block covariance matrix (see Section 2.2.2 and Riihimäki and Vehtari, 2010).

**Operators**   The constraint presented in Eq. 6 can be viewed more generally as an operator $\Phi : \mathcal{U} \to \mathcal{V}$, where $u \in \mathcal{U}$ are the GP latent functions, and $v \in \mathcal{V}$ are the functions that encode the constraints s.t. $v = \varepsilon \, \forall \, v \in \mathcal{V}$. This idea can be used to encode a broad range of constraints, including integral constraints (as also explored in Hansen et al., 2024), and differential operators such as constraints on the divergence or curl of a vector field. Note that the latter can also be achieved by modifying the prior with bespoke covariance kernels (Fuselier, 2006; Baldassarre et al., 2010; Wahlström, 2015; Jidling et al., 2017), but this limits the choice of kernel.

A case of particular interest is when the operator $\Phi$ is a linear map, i.e.

$$\Phi(u(\boldsymbol{x})) = G u(\boldsymbol{x}) - b = \varepsilon \quad \forall \quad \boldsymbol{x} \in \{\tilde{\boldsymbol{x}}_i\}_{i=1}^M. \tag{13}$$

Then the posterior $p_\Phi(\boldsymbol{f} | \boldsymbol{X}, \mathbf{y}, \Phi)$ is exactly a Gaussian with mean and covariance matrix given by Eq. 3. This is an idea that was also explored in the context of physics-informed machine learning by Raissi et al. (2017), who utilise the same identity to jointly model $u$ and $v$. Note that Raissi et al. (2017) present an extension to the work of Graepel (2003) and later Särkkä (2011), and that PDE-constrained cokriging can be traced even further back (Kitanidis and Vomvoris, 1983, and likely further).

## 3   EXPERIMENTS

In this section, we present numerical experiments that demonstrate the QLCR-GP methodology outlined above. In the first example, we require the posterior GP model of a synthetic test function to have a divergence-free gradient. In the second example, we require the posterior heteroskedastic GP model of Thomson scattering data from the MAST tokamak to be both strictly positive and monotonically decreasing. For both experiments, the constraints were imposed using Markov Chain Monte Carlo (MCMC).

### 3.1   DIVERGENCE-FREE GRADIENTS

In this example, we used the modified Branin-Hoo function of Picheny et al. (2013) as the latent data-generating model

$$z(\boldsymbol{x}) = \frac{1}{51.95}\left[\left(\bar{x}_2 - \frac{5.1\bar{x}_1^2}{4\pi^2} + \frac{5\bar{x}_1}{\pi} - 6\right)^2 + \left(10 - \frac{10}{8\pi}\right)\cos(\bar{x}_1) - 44.81\right]$$

$$\text{with}\quad \bar{x}_1 = 15x_1 - 5,\ \bar{x}_2 = 15x_2; \quad (14)$$

We perturbed the raw function values with Gaussian noise, so that the synthetic measurements were

$$\boldsymbol{y} = \boldsymbol{z} + \boldsymbol{\epsilon} \quad \text{with} \quad \boldsymbol{\epsilon} \sim \mathcal{N}(\mathbf{0}, 0.1^2). \quad (15)$$

The unconstrained posterior GP was computed using type-II Maximum Likelihood Estimation (MLE, Rasmussen and Williams, 2006), i.e. by maximising evidence $p(\mathbf{y}|\boldsymbol{X}, \boldsymbol{\theta})$ with respect to the hyperparameters $\boldsymbol{\theta}$. We then conditioned the unconstrained posterior Gaussian distribution on the operator constraint

$$\nabla^2 f(\boldsymbol{x}) = \nabla \cdot \nabla f(\boldsymbol{x}) = \varepsilon \quad \text{with} \quad \varepsilon \sim \mathcal{N}(0, 1^2), \quad (16)$$

requiring the divergence of the gradient to be small. Figure 1 shows the unconstrained GP posterior mean, and the divergence of the gradient of the unconstrained posterior mean, while Figure 2 shows the equivalents for the QLCR-GP posterior. Clearly, the residual divergence of the gradient (Figure 2, right) is non-zero but it has the required distribution $\varepsilon \sim \mathcal{N}(0, 1^2)$

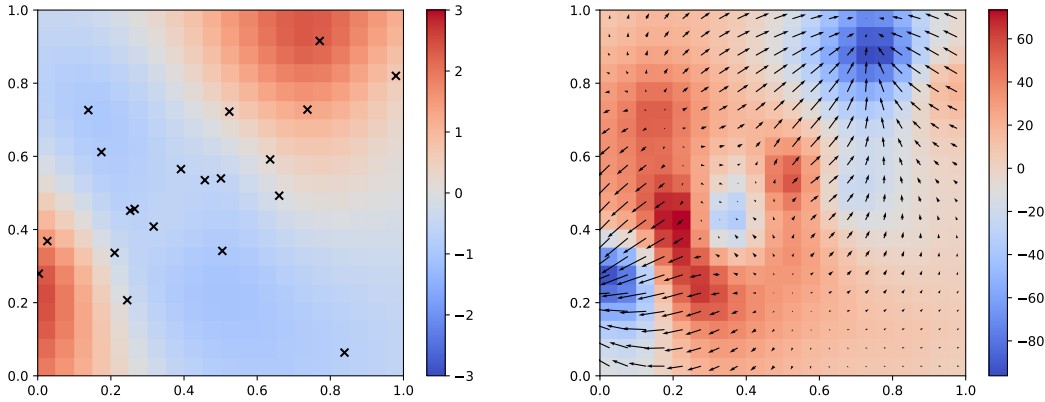

Figure 1: Unconstrained posterior mean $\mu(\mathbf{x})$ and sampling locations (left). The divergence of the gradient of the posterior mean $\nabla^2 \mu(\mathbf{x})$ and the gradient of the posterior mean (right).

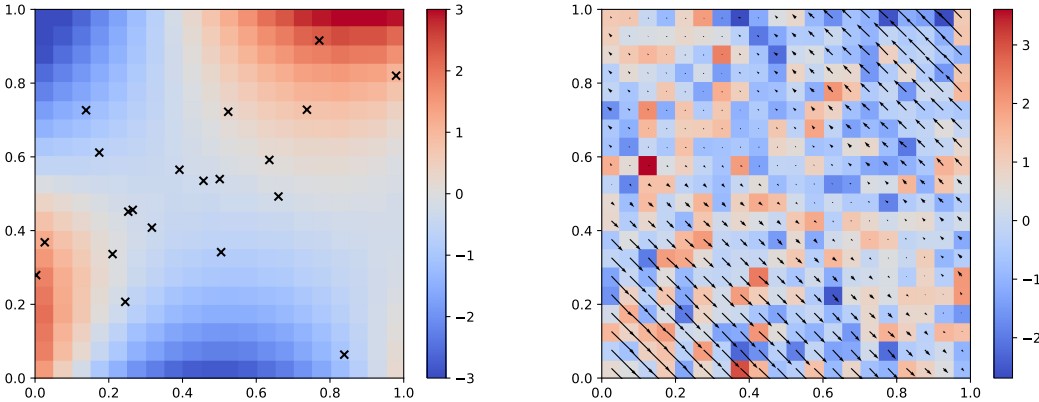

Figure 2: QLCR-GP posterior mean $\mu(\mathbf{x})$ and sampling locations (left). The divergence of the gradient of the QLCR-GP posterior mean $\nabla^2 \mu(\mathbf{x})$ and the gradient of the QLCR-GP posterior mean (right).

## 3.2 DIAGNOSTIC ANALYSIS IN FUSION PLASMA

In fusion devices, fast and accurate measurements of plasma parameters are of critical importance. From the perspective plasma physics, accurate and trustworthy measurements from experiments can further the field's understanding of fundamental principles. Similarly, for the practical operation of a tokamak fusion power-plant, high quality diagnostics can significantly enhance plasma control. A diagnostic critical to both operations and scientific understanding in fusion is Thomson scattering (Evans and Katzenstein, 1969). Thomson scattering is a plasma diagnostic that directs a laser beam at a plasma, and measures the light scattered off free electrons in the plasma. The count of scattered photons can then be be used to infer the plasma density.

Regression of Thomson scattering data from tokamaks is typically executed using a modified hyperbolic tangent (tanh) basis function (Scannell et al., 2006). Though this method allows for the extraction and interpretation of parameters such as the pedestal width and height, it can often by overly prescriptive, inflexible, and, crucially, does not produce profile uncertainties. Hence, recent studies have considered GP regression to fit the scattering data (Chilenski et al., 2015; Kwak et al., 2020). However, the unconstrained GPs employed in these studies are not ideal for Thomson scattering data, which has several underlying constraints imposed by the specific physics. Specifically, the electron temperature $T_e$ and density $\rho_e$ cannot be negative

$$T_e, \rho_e \geq 0. \tag{17}$$

Moreover, since heat and fuel is injected at the centre of the machine, the profiles of these variables will typically be monotonically decreasing from the centre of the core outwards

$$\frac{T_e}{dx}, \frac{\rho_e}{dx} \leq 0. \tag{18}$$

Hence, unconstrained GP regression will typically overestimate the uncertainty of these processes.

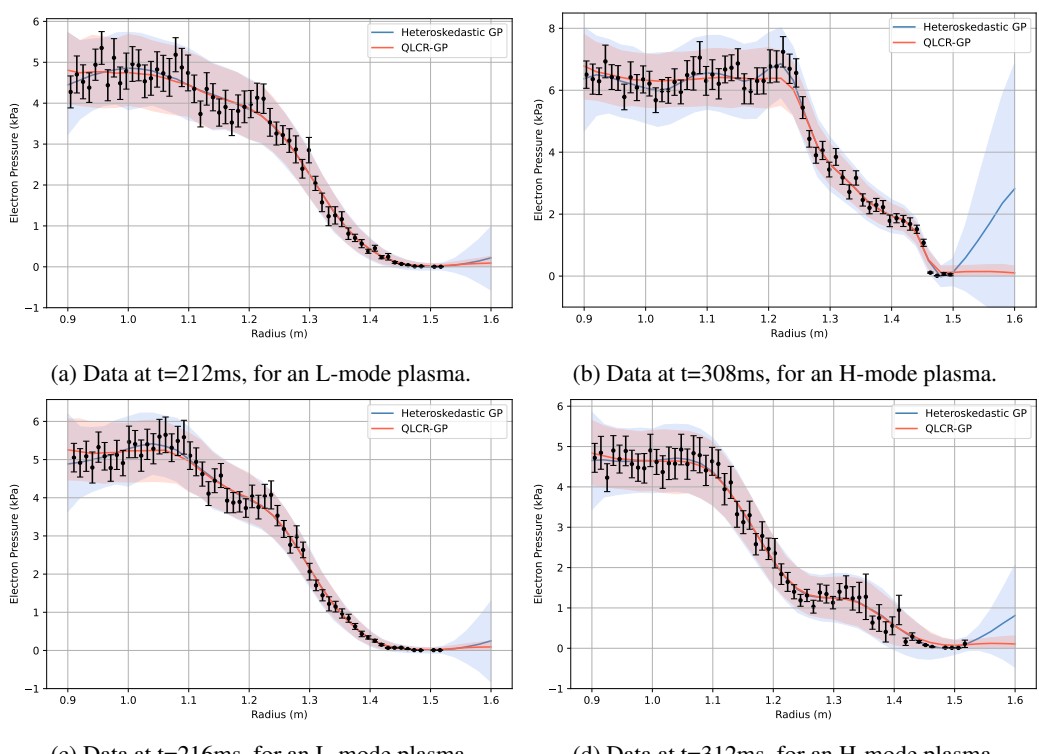

(a) Data at t=212ms, for an L-mode plasma.     (b) Data at t=308ms, for an H-mode plasma.

(c) Data at t=216ms, for an L-mode plasma.     (d) Data at t=312ms, for an H-mode plasma.

Figure 3: A radial profile of Thomson-measured electron pressure in the MAST tokamak shot #27035. Predictions using both constrained and unconstrained GPs are overlaid.

In this study we considered Thomson scattering data from the Mega Ampere Spherical Tokamak (MAST) from shot #27035, which features an L-H transition. We used a GP with a heteroscedastic noise model (Goldberg et al., 1997; Kersting et al., 2007) to model the unconstrained posterior $p(\boldsymbol{f}|\boldsymbol{x}, \boldsymbol{\rho}_e)$. We then reconditioned this distribution on Eq. 6 with constraints according to Eqs. 10 and 12 using Markov Chain Monte Carlo to obtain the $\Phi$-constrained posterior $p_\Phi(\boldsymbol{f}|\boldsymbol{x}, \boldsymbol{\rho}_e, \Phi)$.

As shown in Figure 3, the constrained GPs show improved performance compared to unconstrained GPs for both H and L-mode profiles. Whereas the unconstrained GPs fit undulating curves that dip towards the centre of the machine, the constrained GPs show the physically expected profiles, which are monotonic and flat towards the core. In addition, extrapolation at the edge is both highly uncertain and negative with unconstrained GPs. The physics constrained GPs on the other hand, show highly certain extrapolated profiles that tail off to 0, as is physically expected.

## 4 DISCUSSION

In this paper we have presented Quasi-Likelihood Constraint Relaxation (QLCR-GP), a generalisation of virtual point methods for constrained GP regression. It allows for imposing a broad range of constraints on a GP posterior using virtual measurements of some functional of the GP latent function, which encodes the constraint. A relaxation parameter which is effectively the variance of the constraint residual $\nu^2$, can be viewed as an implausibility measure of the constraint, with which the modeller can encode their belief in the constraint, and tune its importance. We have presented a selection of particular constraints that can be encoded using the approach outlined herein, but any constraint that can be written in the form of Eq. 5 can in principle be handled using the QLCR-GP formalism. This makes the approach a highly flexible one, which can be adapted to a broad range of different regression tasks. However, it does a significant drawback, namely that constraints can only be enforced point-wise (and not globally), which is subject to the curse of dimensionality. We have demonstrated QLCR-GP using two different examples, one of a synthetic function that is required to have a divergence free gradient and one of Thomson scattering data from the MAST tokamak, where the QLCR-GP is required to be both monotonically increasing and strictly positive.

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
