# OpenReview forum: "Constraining Gaussian Processes Regression with Quasi-Likelihood Constraint Relaxation"
_ICLR.cc/2025/Conference — ICLR 2025 Conference Withdrawn Submission_

### Official Review · Reviewer_PYss · 2024-10-16

**Soundness:** 1
**Presentation:** 1
**Contribution:** 1
**Rating:** 1
**Confidence:** 5

**Summary:**

This paper considers the interesting problem of encoding constraints such as monotonocity, convexity, fixed values and positivity to Gaussian processes. This is achieved by adding a number of "virtual points", which are pseudo observations encoding the constraint pointwise. These constraints are only enforced weakly using a Gaussian distribution with mean zero and variance controlled by a parameter nu.

**Strengths:**

The main (and perhaps only) strength of this paper is that it tackles a problem which I personally am interested in.

**Weaknesses:**

This paper is very far from the standard I would expect for an ICLR paper, and is much closer to a workshop paper. Having worked on GPs and published at machine learning conferences for many years, I have rarely seen a submission which is so far from ready to be submitted to a top conference. As a result, my review will be brief and summarise key flaws:

- The language used is very informal and not particularly appropriate for a paper. I would recommend making sure you write more formally and to the point. For example, the introduction reads more like an informal conversation over coffee than an introduction trying to convince the reader of the importance of the problem.

- There is no comparison to existing methods. The authors did review existing work, but then fail to compare against it. How do the advantages/disadvantages of this work show in practice? Do we actually get better approximations than if using competitors? When is it better and when is it worse? How does the computational cost relate?

- There is no discussion of nu. For example, in the example of "fixed values", why would we ever take nu greater than zero? I.e. why not do exact function evaluations, which is straightforward to do with GPs? Is it because this is only feasible for \Phi being linear? This needs to be discussed much more at length. Also, how should we choose \nu for applications? Can this be estimated? What is the impact of estimating?

- There is no discussion of limitations of this method. For example, no mention of the additional cost of using "virtual points", no discussion of the computational cost of approximating the posterior, no discussion of the exact type of algorithm used to approximate the posterior (line 382 mentions MCMC, but which MCMC algorithm? How good was this? How slow was it?).

- A much more minor concern is that I am not sure this paper is very relevant to the ICLR community. Given the AISTATS deadline is so close to that of ICLR, I think submitting the paper to AISTATS would have made much more sense as there would be a much bigger audience that would be interested in this line of work.

Overall, I think having to review this paper has been a waste of valuable reviewing time, and I encourage the authors to consider this when next submitting to a conference.

**Questions:**

I already listed many open questions, and do not have any further questions for the authors. The main reason is that the paper is so far from ready for publication at ICLR that no response from the authors could convince me to change my mind. For full clarity: I am not saying that this topic/line of work is not interesting (it is!), but the authors would need to write a completely new version of the paper to ever convince me that the paper should be accepted.

---

### Official Review · Reviewer_H5rk · 2024-10-28

**Soundness:** 3
**Presentation:** 3
**Contribution:** 2
**Rating:** 3
**Confidence:** 2

**Summary:**

The paper gives a method to incorporate a variety of constraints into Gaussian Processes via reconditioning of an unconstrained Gaussian Process on a quasi-likelihood which encodes the desired constraint. The constraint is relaxed by a residual. If the operator which defines the constraint is linear and the residual is Gaussian, then the likelihood is also Gaussian and the Posterior can be computed in closed form. Otherwise MCMC methods have to be used for posterior approximation.

**Strengths:**

The paper is well written and gives a good overview on the related work

**Weaknesses:**

- The novelty compared to Hansen et al. and Riihimäki and Vehtari seems negligible to me. The authors do not argument improvements of their formulation of constraints compared to existing formulations.
- The experiments only show qualitative results, there is not even one quantitative comparison in this paper.

**Questions:**

- Why is your formulation of constraints better than or fundamentally different from the one of Hansen et al. ?
- Why is your way of incorporating nonlinear constraints better than or fundamentally different from the one of Riihimäki and Vehtari ?
- Is there a difference in runtime to existing methods ?
- What is the background of modeling the fusion reactor ? Is it for real-time or control purposes ? Is your method capable of these real time requirements ?
- What about sparse/variational GP approaches (Titsias 2009, Hensman et al. 2017) for fulfilling the non linear likelihoods ?

---

### Official Review · Reviewer_FgyF · 2024-11-01

**Soundness:** 2
**Presentation:** 2
**Contribution:** 1
**Rating:** 3
**Confidence:** 3

**Summary:**

The paper presents a method for handling constrained Gaussian Processes. The approach uses virtual measurements with a quasi-likelihood function to encode the relevant constraints.

**Strengths:**

The paper tackles an interesting problem for the community: constrained Gaussian Processes. It effectively motivates the issue and includes substantial relevant related work.

**Weaknesses:**

- The methods section lacks an in-depth analysis of the proposed method. For example, there is no discussion about the resulting posterior; the authors merely state that "the posterior can be approximated using variational methods or Markov Chain Monte Carlo," but this should be explored in more detail. They could, for instance, present the Evidence Lower Bound (ELBO) for the variational inference approach. Additionally, the authors should discuss the method's limitations and examine the implications of different choices for $\Phi$.

- The weakest part of the paper is the experiments section. The authors do not compare their method with other relevant approaches; the only comparison is against a heteroscedastic Gaussian Process. This seems unusual since this model does not handle the constraints addressed in the experiment. The authors should consider comparing their method with approaches such as Riihimäki and Vehtari (2010).

- Overall, I find it challenging to identify the main contribution of this work. Although the authors claim it is a framework for handling general constraints, they do not explore specific approaches to address the potential intractability of the posterior. The method essentially provides an alternative perspective on constraints without offering clear benefits. In contrast, previous approaches, tailored to specific constraints, have more effective ways of managing posterior intractability.

**Questions:**

- How does $\nu$ selected in practice? How sensitive is the method to the choice of $\nu$?

---

### Official Review · Reviewer_8cwR · 2024-11-03

**Soundness:** 1
**Presentation:** 2
**Contribution:** 2
**Rating:** 3
**Confidence:** 5

**Summary:**

This work considers the problem of imposing constraints on the paths of Gaussian Processes (GPs), including path values, derivatives, and general operators. To do so, it penalize the likelihood of GPs in order to induce the desired properties, described as general operator equations. The proposed method is showcased in low-dimensional scientific application.

**Strengths:**

- The problem is well-motivated, timely, and sufficiently well-positioned in the literature
- The proposed method appears straightforward to apply and seems to be effective in the proposed experiments
- The paper is clearly written and organized

**Weaknesses:**

While the topic is interesting and timely, the current version of this manuscript has major weaknesses related to (1) limited description, missing details; (2) limited experimental validation; (3) applicability; (4) theoretical grounding and guarantees.

1. **Limited descriptions, missing details:** while the manuscript is well-written, it leaves many details of the proposed method to be filled in by the reader. This makes it difficult to judge its applicability in complex scenarios. For instance, it does not explicit state any posteriors except for (3)-(4), which is a particular case of the proposed method already put forward in [Hansen et al., 2024]. It is therefore hard to judge the novelty of the proposed method. Additionally, without explicit forms for the posteriors, it is hard to judge the computational complexity of the proposed approach, especially since no details are provided on the applications beyond the fact that they were sampled using MCMC. It is therefore unclear which MCMC technique is used (particularly since it must sample in function spaces), how the presence of constraints affects the convergence of these methods, and what effect does the specific MCMC method used has on how well the constraints are enforced. This particularly important since the advantage of GPs are their closed-form posterior that can be leveraged to reduce the computational complexity of inference. If MCMC methods are deployed, it is unclear why the retriction to GPs is important. The manuscript should include a more detailed discussion of all these aspects before it is possible to judge its contribution.

1. **Limited experimental validation:** The issue in (1) is exacerbated by the limited experimental validation. The applications considered are limited to low-dimensional settings with loose, linear constraints. This does not properly showcase the performance or limitations of the approach. In the first application, the "constraint component" ($\sigma^2 = 1$) in the posterior is substantially more lax than the "data component" ($\sigma^2 = 0.01$). Hence, while it modifies the solution (Fig. 2), it does not enforce the "divergence-free" constraint in (16). Doing so would require using a very small $\nu$, which would in turn affect the numerical stability of the MCMC method (see (4) below). Including additional experiments or theoretical results on the effects of $\nu$ is necessary to support the use of this method to *enforce* constraints. Even if this is not the goal of the paper, i.e., it seeks only to induce certain behaviors without guaranteeing them, it is still unclear to what extent $\nu$ affect *how much* that behavior is induced or how it interacts with other terms in the posterior. I cannot judge the second application as it lacks details on the choices of parameters altogether, but the effect of adding the constraint does not appear to be significant (Figure 3). Once again, this could be due to the data "taking over" the posterior, but it is impossible to tell. Finally, since the constraints in both applications are linear, it would appear that other baselines could also be considered. For instance, using virtual points to fix values and techniques from Section 2.2.2 or 2.2.3 (i.e., [Hansen et al., 2024] or (3)-(4)). Leaving out these comparisons also makes it difficult to judge the novelty of the method compared to the previous literature (see (1) above).

1. **Limited applicability:** The last point in (2) is particularly critical as the proposed method does not *enforce* constraints, but rather penalizes its violations. As such, the resulting inference is not representative of the constrained Bayesian problem. Substantial theoretical and experimental validation is required to showcase to what extent constraints are actually *enforced* (depending on $\nu$). Additionally, the curse of dimensionality issue related to virtual measurements (raised in line 129) is not tackled by this method (that itself relies on virtual points as per line 112). This limitation, however, is only discussed in the context of related work.

1. **Theoretical grounding and guarantees**: Since the manuscript relies primarily on MCMC methods for inference, it should more carefully examine that literature, particularly since penalties in MCMC are not commonly used in a myriad of context (including inducing constraints), see, e.g., [Robert and Casella, "Monte Carlo statistical methods"]; [Karagulyan et al., "Penalized Langevin dynamics with vanishing penalty for smooth and log-concave targets," NeurIPS'20]; [Gurbuzbalaban et al., "Penalized Overdamped and Underdamped Langevin Monte Carlo Algorithms for Constrained Sampling," JMLR'24]. While these methods do not target function spaces, they can still be used to sample paths from GPs. In the particular case of linear constraints, they can even be used to sample constrained paths (see [Pfortner et al., "Physics-Informed Gaussian Process Regression Generalizes Linear PDE Solvers," arxiv'22]).
This last point raises another critical issue with the proposed method. Note that (3)-(4) are only valid for bounded linear operators under particular assumptions (see Thm. 1 and Cor. 2 in [Pfortner et al., 2022]). These are quite mild, but should still be mentioned in the paper. What is more, it is not clear to what extent this generalizes to arbitrary operators as in (5). In general, conditioning a GP on (5) is ill-posed and may lead to singular paths among other measurability issues. It is therefore unclear whether (7) makes sense without strong assumption on $\Phi$ (at the very least to guarantee the existence of the integral $p(\Phi \mid X, y)$).

In view of these issues, I cannot recommend the publication of the current version of this manuscript.

**Questions:**

See Weaknesses above.

---

### Note · Authors · 2024-11-25

**Comment:**

We would like to thank the reviewers for their time.

Considering their comments we believe that the paper requires more work before publication.

**Withdrawal Confirmation:**

I have read and agree with the venue's withdrawal policy on behalf of myself and my co-authors.